# MHFS-FORMER: Multiple-Scale Hybrid Features Transformer for Lane Detection

**DOI:** 10.3390/s25092876

**Published:** 2025-05-02

**Authors:** Dongqi Yan, Tao Zhang

**Affiliations:** Faculty of Information Engineering and Automation, Kunming University of Science and Technology, Kunming 650500, China; yandongqi@stu.kust.edu.cn

**Keywords:** lane detection, vision transformer, feature fusion

## Abstract

Although deep learning has exhibited remarkable performance in lane detection, lane detection remains challenging in complex scenarios, including those with damaged lane markings, obstructions, and insufficient lighting. Furthermore, a significant drawback of most existing lane-detection algorithms lies in their reliance on complex post-processing and strong prior knowledge. Inspired by the DETR architecture, we propose an end-to-end Transformer-based model, MHFS-FORMER, to resolve these issues. To tackle the interference with lane detection in complex scenarios, we have designed MHFNet. It fuses multi-scale features with the Transformer Encoder to obtain enhanced multi-scale features. These enhanced multi-scale features are then fed into the Transformer Decoder. A novel multi-reference deformable attention module is introduced to disperse the attention around the objects to enhance the model’s representation ability during the training process and better capture the elongated structure of lanes and the global environment. We also designed ShuffleLaneNet, which meticulously explores the channel and spatial information of multi-scale lane features, significantly improving the accuracy of target recognition. Our method has achieved an accuracy score of 96.88%, a real-time processing speed of 87 fps on the TuSimple dataset, and an F1 score of 77.38% on the CULane dataset. Compared with the methods based on CNN and those based on Transformer, our method has demonstrated excellent performance.

## 1. Introduction

In the current era of the vigorous development of intelligent driving, the significance of lane detection has become increasingly prominent. As a crucial perception component in autonomous driving and Advanced Driver Assistance Systems (ADAS), it offers support for the precise navigation and path planning of vehicles. Lane-detection technology is generally employed to accurately locate the shape of each lane within traffic scenarios. As depicted in Figure 1, lane detection confronts numerous challenges in real-world traffic scenarios. This is because lanes usually feature complex layouts, such as dense, curved, and bifurcated structures. They are also subject to interference from environmental lighting. In extreme lighting conditions like glare and low light, the line of sight is disrupted, thereby affecting the detection accuracy. Moreover, moving vehicles and pedestrians passing through can obstruct the lane, disrupting their continuity, thus presenting a multitude of difficulties.

Traditional lane-detection methods rely on hand-designed models [1,2,3] to extract lane features and the traditional Hough transform method [1] to match lanes. There are also methods of designing template matching [2,3]. According to prior knowledge, such as the shape and width of lane lines, some templates of lane lines are designed, and then the regions similar to the templates in the image are searched for as lanes. Methods of this kind have weak generalization ability and can only be used to detect lanes in specific scenarios, making it difficult to apply them in complex scenarios. However, benefiting from the continuous advancements in deep learning and computer vision, lane detection based on neural networks has achieved excellent performance and significantly enhanced the generalization ability, enabling it to deal effectively with complex real-world scenarios. Some studies utilize semantic segmentation methods [4,5,6,7,8,9], which adopt a pixel-level prediction approach to segment the lane pixels in an image. Then, clustering methods are employed to classify the segmented lane pixels into different instances. This has improved the drawback of poor generalization ability in traditional methods. However, the non-end-to-end manner incurs a large overhead, making semantic segmentation methods lack real-time detection capability. On the other hand, anchor box-based methods [10,11,12,13] achieve the detection of lanes by reasonably defining line anchors or row anchor points. Currently, row anchor point-based lane-detection methods demonstrate excellent inference speed at the expense of some accuracy and have become the preferred solution for addressing real-time constraints. In addition, parameter regression-based methods utilize neural networks to perform regression prediction on a set of parameters and then construct a curve-fitting model based on these parameters to detect lanes. This method effectively reduces the number of parameters and achieves a good balance between performance and real-time performance. For example, in PolyLaneNet [14], lanes are assumed to be polynomial parameters. However, due to its neglect of learning global information, its performance lags behind that of well-designed segmentation methods.

To address these challenges, recently, the DETR [15] method based on the Transformer architecture has been adopted for lane detection. What makes lane detection special compared to general image vision tasks is that lanes often span the entire image instead of being concentrated in a certain sub-region, and lanes have the characteristic of being long and slender in structure, which requires the use of a relatively refined feature detector for detection. Therefore, the lane-detection model using the Transformer model of the DETR architecture not only makes use of the basic underlying features extracted by the CNN backbone network but also integrates the advantages of the Transformer’s self-attention mechanism [16]. It can excavate the deep-seated and latent semantic information in the data, thereby enhancing the model’s ability to understand complex content. Moreover, it utilizes bipartite graph matching to train predictions and labels.

During the testing process, it can directly generate all the results, thus avoiding the complex post-processing procedures. LSTR [17] has been proposed as an end-to-end method, which uses Transformer blocks to directly learn the curve parameters of the lane shape model through a network established based on DETR. O2SFormer [18] has been proposed as a model that introduces a one-to-many label assignment strategy to address the issue in the lane recognition process where the label semantics reduce the training efficiency in DETR. However, these methods do not make full use of the features extracted by the backbone network. The DETR architecture has the drawbacks of excessive computational overhead, which makes it unsuitable for adding multi-scale features, and it also performs poorly in recognizing small structures. Recent approaches have addressed these shortcomings and brought about performance improvements. Deformable-DETR [19] has been innovatively designed as a deformable attention module. This module focuses on a small number of sampled positions and serves as a pre-filter to screen out prominent key elements from all the pixels of the feature maps. It combines the advantages of the sparse spatial sampling of deformable convolutions and the relational modeling ability of the Transformer. Without sacrificing performance while reducing the number of parameters, it accelerates the convergence of the model. Moreover, this module can be naturally extended to aggregate multi-scale features, thus solving the problem of DETR’s poor performance in recognizing small structures. RT-DETR [20], on the other hand, has been designed an efficient hybrid encoder. By decoupling the intra-scale interaction and cross-scale fusion, it can process multi-scale features rapidly to improve the speed. Additionally, it proposed a query selection method with the least uncertainty, which provides high-quality initial queries for the decoder, thereby enhancing the accuracy. These improvements have made detectors comparable to DETR more competitive.

Based on the above analysis, we propose an end-to-end model named MHFS-FORMER based on the Transformer to address these issues. MHFS-FORMER takes multi-scale features as input information and introduces a novel multi-reference deformable attention module that can disperse the attention around the objects. This can accelerate the convergence and enhance the model’s representation ability during the training process, enabling it to better capture the elongated structures of the lanes and the global environment. We utilize MHFNet to enhance the mixed multi-scale features. The multi-layer features extracted by the backbone network are cascaded and fused with the output of the Transformer Encoder. By combining the high-level features in the early stages to guide the learning of low-level features in subsequent stages, the effectiveness of multi-scale feature fusion is enhanced. Moreover, we have also designed ShuffleLaneNet, which conducts a meticulous exploration of the channel and spatial information in the lane features. Through enhancing feature extraction, the accuracy of target recognition has been significantly improved. We have conducted experiments on the widely used lane datasets TuSimple [21] and CULane [4]. On the TuSimple dataset [21], we have achieved an accuracy rate of 96.88% and a real-time processing speed of 87 fps. On the CULane dataset [4], we have obtained an F1 score of 77.38%, which also demonstrates state-of-the-art performance among the lane-detection methods based on parameter regression.

Our contributions are summarized as follows:We have proposed a novel end-to-end Transformer-based lane recognition model, MHFS-FORMER. It introduces enhanced multi-scale mixed features into the Transformer model, captures the elongated structures of lanes and the global environment through the multi-reference deformable attention module, and eliminates the complex post-processing procedure at the same time;We propose a multi-scale fused feature network rich in semantic information named MHFNet. It is initiated by fusing two adjacent high-level features. Multiple cascaded stages are combined to iteratively extract and fuse multi-scale features. Moreover, considering the potential information conflicts that may occur during the feature fusion process at each spatial position, adaptive spatial fusion operations are further employed to mitigate these inconsistencies;We have designed a network dedicated to the feature mining of lane lines named ShuffleLaneNet. Exploring the channel information and spatial information enhances the lane features. Based on maintaining the number of model parameters, it significantly improves the performance of the model;The proposed MHFS-FORMER achieves encouraging performance on the TuSimple and CULane datasets. (That is, it attains an F1 score of 77.38% on CULane, an accuracy score of 96.88%, a real-time processing speed of 87 fps, and a false positive rate (FPR) of 2.34% on TuSimple.).

## 2. Related Work

This section briefly introduces the deep learning-based lane-detection methods and the approaches of Vision Transformer.

### 2.1. Lane Detection

With the development of deep learning, an increasing number of methods based on deep neural networks have demonstrated superior performance in the field of lane detection. The lane-detection frameworks in deep learning can be classified into four main types: methods based on segmentation, methods based on anchor boxes, methods based on parameters, and methods based on keypoints detection.

#### 2.1.1. Methods Based on Segmentation

The segmentation-based methods [4,5,6,7,8,9] adopt the approach of pixel-level classification prediction to separate the lane regions from the background so as to obtain the lane pixels in the image. SCNN [4] utilizes the message-passing mechanism to improve the spatial relationships in the semantic segmentation framework for lane detection. However, its message-passing mechanism accomplishes the segmentation prediction by recursively aggregating spatial information, which leads to high latency. As a result, it is difficult for SCNN [4] to achieve real-time detection. Moreover, it also requires assuming that the number of lane lines is fixed (for example, 4 lines) in rows. LaneNet [5] regards each lane as an instance and predicts the lane lines through an instance segmentation model. This breaks through the limitation of only being able to predict a fixed number of lane lines, enabling the model to better handle a variable number of lines using the instance segmentation pipeline. However, it requires post-inference clustering to generate line instances. RESA [6] makes use of the shape information of the lane and manually designs a real-time feature transmission aggregator to capture the spatial relationships between non-adjacent pixels. Nevertheless, due to its pixel-level processing, the computational cost remains high. SAD [7] sets up a self-attention distillation mechanism between each feature encoding stage, achieving the lightweighting of the network. CondLaneNet [8], aiming at the problem of lane instance-level differentiation, proposes a conditional lane-detection strategy based on conditional convolution and row-by-row representation. It first detects the starting points of the lanes, dynamically generates kernels according to the differences of the starting points, and detects the lanes through the convolution of the kernels with the entire feature map. LaneAF [9] uses a method that combines the Affinity Field with binary classification segmentation for lane detection and instance segmentation. It clusters the lane pixels into corresponding lane instances in both the horizontal and vertical directions and can detect lanes with a varying number.

#### 2.1.2. Methods Based on Anchor Boxes

The anchor box-based methods [10,11,12,13] use predefined anchors to identify the potential locations of lane markings. They focus on classifying and regressing the anchor boxes that are likely to contain lane segments. Compared with the segmentation methods, this usually reduces the computation and can improve the real-time performance. Line-CNN [10] was the first method to propose the line anchor approach. Subsequently, LaneATT [11] added line anchors with an attention mechanism to aggregate global information, improving both accuracy and efficiency. This work introduced a novel anchor-based attention mechanism. CLRNet [12] proposed a cross-layer optimized lane-detection network. This network uses high-level features to detect lanes and low-level features to refine the positions of the lane, making use of more contextual information for lane detection. On the other hand, UFLD [13] designed a novel row anchor detection method, which offers the advantages of simplicity and speed for lane detection. By dividing the pixels on the predefined row anchors of the image into several position grids and then using a fully connected layer to regress and predict the position grids to which the lane positions on each predefined row anchor belong, it greatly reduces the computational cost of the forward inference of the network. This method sacrifices some accuracy but demonstrates excellent inference speed and has become the preferred solution for addressing real-time constraints.

#### 2.1.3. Methods Based on Parameter Regression

The parameter regression methods regard the lane as a complete curve and use a concise curve (such as a cubic function, Bezier curve, etc.) to describe a lane, transforming the lane-detection task into a regression task of the curve parameters. PolyLaneNet [14] describes the lane with a cubic polynomial and directly regresses the parameters of the cubic function through the backbone network, achieving a huge improvement in real-time performance. LSTR [17] uses a polynomial model to represent a lane and innovatively applies the Vision Transformer model to the lane-detection task. This method uses the DETR model architecture, which enables it to perceive global information more accurately and make more accurate predictions of the polynomial coefficients. BezierLaneNet [22] introduces the Bezier curve, represents the lane as a third-order Bezier curve, and converts the prediction of the polynomial coefficients into the detection of specific control points.

#### 2.1.4. Methods Based on Keypoints

The methods based on keypoint detection regard the lane detection as a task of detecting a series of keypoints on the lane. The target detection model is used to obtain several keypoints of the lane, and these keypoints are interpolated to finally estimate the position of the lane. PINet [23] uses an advanced stacked hourglass network and a clustering algorithm, achieving good detection results. However, it requires intensive post-processing, which increases the computational load. FOLOLane [24] adopts a heatmap-based keypoint detection method. It infers the keypoints on the lane sequentially from bottom to top through the local information of the image. This refinement of the positions of the keypoints within the local range has achieved high accuracy.

### 2.2. Vision Transformer

Transformer [16], an encoder-decoder model based on the self-attention mechanism, demonstrates great strength in modeling global context and has revolutionized the field of Natural Language Processing (NLP). Inspired by these significant achievements, in recent years, there have been some pioneering works in the field of Computer Vision (CV) that adopt architectures similar to the Transformer to improve visual tasks. These architectures outperform CNN-based backbone networks in many visual tasks. Carion et al. proposed DETR [15] based on the Transformer. They use a CNN backbone network to extract compact feature representations and then utilize the Transformer Encoder–decoder and a simple feed-forward network (FFN) to carry out the final object detection task. DETR treats object detection as a set prediction problem, simplifying the overall process of object detection. By leveraging the powerful relational modeling ability of the Transformer blocks, it eliminates the need for hand-crafted components and post-processing steps such as non-maximum suppression. Based on the relationships between objects and the global context, it directly outputs the final prediction set in parallel, achieving an end-to-end model architecture. Inspired by the interesting design and excellent performance of DETR, Zhu et al. developed Deformable-DETR [19]. Drawing on the idea of deformable convolution, which processes image feature maps through a sparse sampling and deformable attention mechanism, they proposed a deformable attention module in combination with the long-range relationship modeling of the Transformer. This module only focuses on the prominent key elements in all feature maps and extends a single feature to a multi-scale hierarchy. RT-DETR [20] has significantly improved the inference speed by designing an efficient hybrid encoder that decouples the intra-scale interactions and cross-scale fusion of features at different scales. Moreover, it has devised a query selection method with minimal uncertainty, which supplies high-quality initial queries to the decoder, thereby enhancing the accuracy. Conditional-DETR [25] restricts the object search scope by introducing a Conditional Spatial Query. By merely modifying the cross-attention layer, it has achieved a 6 to 10 times improvement in training speed. This approach decouples spatial information from content information, reducing the reliance on high-quality content embeddings and thereby accelerating the convergence rate. These enhancements render detectors with architectures similar to DETR more competitive.

## 3. Structure

This section is constructed based on the geometric prior features of the lane and the global information of images, and MHFS-FORMER is designed. By enhancing features at multiple scales, it improves the recognition effect of the Transformer model on slender structures, while reducing the computational complexity.

### 3.1. Network Overview

The MHFS-FORMER shown in Figure 2 consists of a backbone, ShuffleLaneNet, a Transformer Encoder block, MHFNet, and a Transformer Decoder block.

### 3.2. Lane Parameter Model

In MHFS-FORMER, the prior model of the lane shape is defined as a polynomial structure on the road:(1)$$X=\sum_{n=0}^{N}a_{n}Y^{n}+b,$$where (X,Y) represents the points of the lane on the ground, N is the order of the polynomial X, $a_{n}$ and b are the coefficients of the lane polynomial function. Generally, the shape of lanes in real driving environments is not particularly complex. Therefore, a cubic curve is used to approximate a single lane line on a flat ground, and it can thus be expressed as:(2)$$X=kZ^{3}+mZ^{3}+nZ+b,$$where k, m, n and b are real number parameters, and k≠0. (X,Z) indicates the points on the ground plane. 

In a real driving environment, the road surface may be uneven, which will interfere with the polynomial model. Therefore, it is considered to transform the cubic polynomial of a flat road into a bird’s-eye view for representation to avoid the influence caused by the uneven road surface. The transformation formula is as follows:(3)$$\left\{\begin{array}{l} X=x\times f_{x}\times Y\\ Y=\frac{H}{y\times f_{y}} \end{array}\right\},$$where (X,Y) is the corresponding pixel point on the transformed phase plane, $f_{x}$ is the width of a pixel on the focal plane divided by the focal length, $f_{y}$ is the height of a pixel on the focal plane divided by the focal length, and H is the mounting height of the camera. Considering that there will be bumps during driving, which may cause the mounted camera to tilt. Suppose the tilt angle is θ, the transformation between the non-tilted and tilted cases is as follows:(4)$$\left\{\begin{array}{l} x=x^{\prime }\\ \left[\begin{array}{c} f\\ y \end{array}\right]=\left[\begin{array}{cc} \cos\theta & \sin\theta \\ -\sin\theta & \cos\theta \end{array}\right]\left[\begin{array}{c} f^{\prime }\\ y^{\prime } \end{array}\right] \end{array}\right\},$$where f is the camera focal length, $f^{\prime }$ is the camera focal length after the pitch transformation, and $\left(x^{\prime },y^{\prime }\right)$ is the corresponding pixel point after the pitch transformation. By combining Equations (2) to (4), we can obtain:(5)$$x^{\prime }=\frac{k^{\prime }}{{\left(y^{\prime }-f^{\prime \prime }\right)}^{2}}+\frac{m^{\prime }}{y^{\prime }-f^{\prime \prime }}+n^{\prime }+b^{\prime }\times y^{\prime }-b^{\prime \prime },$$where $f^{\prime \prime }=f\sin\theta$, $k^{\prime }=\frac{k\cos^{2}\theta \times H^{2}}{f_{x}\times f_{y}^{2}}$, $m^{\prime }=\frac{m\cos^{2}\theta \times H}{f_{x}\times f_{y}}$, $n^{\prime }=\frac{n}{f_{x}}$, $b^{\prime }=\frac{b\times f_{y}}{f_{x}\cos\theta \times H}$, $b^{\prime \prime }=\frac{b\times f_{y}\times f\tan\theta }{f_{x}\times H}$. In addition, this model also takes into account the confidence of the lane (0 represents the background and 1 represents the lane marking). In summary, the output of the lane line numbered i is parameterized as:(6)$$p_{i}=(c,k^{\prime },m^{\prime },n^{\prime },f^{\prime \prime },b^{\prime },b^{\prime \prime }),$$where i∈{0,1,…,M}, and M is the total number of lanes.

### 3.3. MHFNet

Considering that the lane-detection task is difficult to perform in complex scenarios, we introduce a multi-scale feature extraction and fusion network into the Transformer model. The multi-layer features extracted by the backbone network are fused in a cascading waterfall manner. By combining the low-level features in the early stages to guide the learning of high-level features in subsequent stages, the effectiveness of multi-scale feature fusion is enhanced. However, current popular Vision Transformer methods usually employ single-scale feature detection to reduce the excessively high computational burden. Alternatively, they directly introduce multi-scale features into the Transformer model by reducing the computational complexity of the self-attention mechanism. Since these methods allocate most of the computational resources to the Transformer Block, the feature fusion in these networks occurs relatively late, which may lead to insufficient feature fusion. Inspired by RT-DETR [20] and AFPN [26], we propose a multi-scale fused feature network rich in semantic information, referred to as MHFNet. It initiates by fusing two adjacent high-level features, incorporates multiple cascading stages to iteratively extract and fuse multi-scale features, and gradually adds low-level features to the fusion process. This makes the semantic information of features at different levels more closely related, enhances the entire feature hierarchy with the rich semantic information of low-level features, and achieves a lightweight module design. Moreover, considering the potential information conflicts that may occur during the feature fusion process at each spatial position, we further utilize an adaptive spatial fusion operation to mitigate these inconsistencies.

Figure 3 illustrates the overall architecture of MHFNet. We use the features output from the last residual block of each stage of ResNet as the multi-scale inputs for the Transformer Block. For the outputs of conv 3, conv 4, and conv 5, we represent the outputs of these last residual blocks as $\left\{C_{3},C_{4},C_{5}\right\}$, and note that they have strides of {8, 16, 32} pixels relative to the input image. We unify the number of channels of these features to 256 through a 1 × 1 convolutional layer. Since conv 1 and conv 2 have a large memory footprint, we do not include them in the feature pyramid. This final set of feature maps is denoted as $\left\{S_{3},S_{4},S_{5}\right\}$, corresponding to spatial sizes of H/8 × W/8, H/16 × W/16, and H/32 × W/32, respectively. $F_{5}$, which is obtained by calculating $S_{5}$ through the single-scale encoder, along with $S_{3}$ and $S_{4}$, are used as the inputs to MHFNet and are represented as $\left\{S_{3},S_{4},F_{5}\right\}$.

Since the encoder-decoder of the Transformer Block uses features with a unified number of channels, we fix the feature dimension (the number of channels, denoted as d) in all feature maps. In this paper, we set d = 256, so all additional convolutional layers have an output of 256 channels.

First, we fuse the two adjacent high-level features, S4 and F5, through the fusion module to obtain the fused features P4 and P5 at their respective scales. The fusion block contains two 1 × 1 convolutions to adjust the number of channels, and N RepBlocks composed of RepConv [27] are used for feature fusion. The outputs of the two paths are fused by element-wise addition. One part of the features extracts semantic information through multiple convolutional layers, while the other part directly retains the original information. Then, these two parts of features are fused by addition so that the fused features contain both the abstracted high-level features and the original low-level detail features. The structure is shown in Figure 4.

In addition, to align the dimensions and prepare for feature fusion, we use 1 × 1 convolutions and bilinear interpolation to upsample the features. On the other hand, according to the required downsampling rate, we use different convolutional kernels and strides to perform downsampling. We apply a 2 × 2 convolution with a stride of 2 to achieve 2-times downsampling and a 4 × 4 convolution with a stride of 4 to achieve 4-times downsampling.

Then, we fuse the features of the three levels, namely S3, P4, and P5. We use ASF (Adaptive Spatial Fusion) to assign different spatial weights to features at different levels during the multi-layer feature fusion process, strengthening the expression of key information, reducing the interference between features of different objects, and improving the fusion effect. We fuse the features of the three levels. Let $x_{ij}^{n\to l}$ denote the feature vector at position (i,j) from level n to level l. The resulting feature vector, denoted as $y_{ij}^{l}$, is obtained through the adaptive spatial fusion of multi-level features and is defined as a linear combination of the feature vectors $x_{ij}^{1\to l}$, $x_{ij}^{2\to l}$ and $x_{ij}^{3\to l}$ as follows:(7)$$y_{ij}^{l}=\alpha_{ij}^{l}\cdot x_{ij}^{1\to l}+\beta_{ij}^{l}\cdot x_{ij}^{2\to l}+\gamma_{ij}^{l}\cdot x_{ij}^{3\to l},$$where $\alpha_{ij}^{l}$, $\beta_{ij}^{l}$ and $\gamma_{ij}^{l}$ represent the spatial weights of the features of the three levels at position l, and they need to satisfy the constraint condition $\alpha_{ij}^{l}+\beta_{ij}^{l}+\gamma_{ij}^{l}=1$. Finally, through the concat operation, we obtain the mixed multi-scale features $\left\{P_{3}^{\prime },P_{4}^{\prime },P_{5}^{\prime }\right\}$, and their feature maps are feature maps with scales of H/16 × W/16, H/32 × W/32 and H/64 × W/64, respectively, and the number of channels is 256.

### 3.4. ShuffleLaneNet

Due to the narrow structure of the lane, there is relatively little useful information in the images. To enhance the accuracy of lane positioning, inspired by [28,29], we propose a ShuffleLaneNet that is used between the backbone and the Transformer Encoder. Although CBAM [28] can help the model pay better attention to the regions where the target objects are located and the important feature channels related to the target objects, it has good versatility and scalability. However, in its design, it mainly focuses on the attention mechanisms in the two dimensions of feature channels and spatial positions while ignoring the interactions between features and so on. Therefore, in some complex computer vision tasks, CBAM [28] may not fully meet the model’s requirements for feature extraction and processing. SA [29] divides the input feature map into multiple sub-feature groups along the channel dimension and simultaneously constructs channel attention and spatial attention for each sub-feature group to highlight the correct semantic feature regions. After processing, all sub-features are re-aggregated, and then a “channel shuffle” operation is adopted to achieve information exchange between different sub-features. However, the information flow between different groups leads to the insufficient exploration of the information of spatial and notification attention. Therefore, in this study, we propose ShuffleLaneNet, as shown in Figure 5. This mechanism fully considers the information flow between feature channels and spatial positions and effectively fuses the interactions between these two types of features. It not only maintains the light weight of the model but also significantly improves the performance of the model. Finally, ShuffleNet units are used to achieve the cross-group information flow along the channel dimension. The channel and spatial information in the lane features are carefully explored to enhance the important information and provide more valuable information for the subsequent modules.

This algorithm takes the feature map $X\in R^{C\times H\times W}$ as the input, where C, H and W represent the number of channels, the spatial height, and the width, respectively. ShuffleLaneNet first divides X into two groups along the channel dimension. The first group conducts spatial attention calculation and channel attention fusion calculation on the feature map, and its feature is $X_{1}\in R^{C\times H\times W}$. The second group is further divided into two groups along the channel dimension. One group focuses on spatial attention calculation and its feature is $X_{2}\in R^{C/4\times H\times W}$, while the other group focuses on channel attention calculation and its feature is $X_{3}\in R^{C/4\times H\times W}$. Finally, they are merged back to the original dimension according to the number of channels, and a channel shuffle operation is adopted to suppress the noise. In the channel attention, we compress the global spatial information into the channel descriptors. Using global average pooling, we generate the channel statistics $c_{k}\in R^{C/n\times 1\times 1}$. Let $X_{1}$ be the result of the first pooling operation with *n* = 2 and $X_{3}$ be the result of the second pooling operation with *n* = 4. Formally, they can be calculated by reducing H×W along the spatial dimension X as follows:(8)$$y_{ij}^{l}=\alpha_{ij}^{l}\cdot x_{ij}^{1\to l}+\beta_{ij}^{l}\cdot x_{ij}^{2\to l}+\gamma_{ij}^{l}\cdot x_{ij}^{3\to l},$$

In addition, a compact function is created to guide the precise and adaptive selection. This is achieved through a simple gating mechanism with a Sigmoid activation function. Then, the final output of the channel attention can be obtained as follows:(9)$$X_{k}^{\prime }=s(F_{c}(c))\cdot X_{k}=s(W_{1}s+b_{1})\cdot X_{k},$$

$W_{1}\in R^{C/n\times 1\times 1}$ and $b_{1}\in R^{C/n\times 1\times 1}$ are the parameters used for scaling and shifting *s*.

We generate spatial attention by leveraging the spatial relationships of the features. To calculate the spatial attention, first, channel pooling is performed on the feature map. We use Group Normalization (GN) [25] for $X_{1}$ and $X_{2}$ to obtain the spatial statistics. For $X_{1}$, *n* = 2, and for $X_{2}$, *n* = 4. Then, the obtained vectors are activated to obtain the attention weights in the spatial dimension and $F_{c}(\cdot )$ is adopted to enhance the representation of $X_{m}$. This method is used to associate long-term dependencies. The final output of the spatial attention is obtained as follows:(10)$$X_{m}^{\prime }=\sigma (W_{2}\cdot GN(X_{m})+b_{2})\cdot X_{m},$$

$W_{2}\in R^{C/n\times 1\times 1}$ and $b_{2}\in R^{C/n\times 1\times 1}$.

After that, all the sub-features are aggregated. We have $X_{2}^{\prime }$ and $X_{3}^{\prime }$. Finally, they are concatenated and then connected with $X_{1}^{\prime }$. We adopt the ShuffleNet unit to achieve cross-group information flow along the channel dimension. The final output of ShuffleLaneNet has the same size as X.

### 3.5. Transformer Block

The Transformer architecture shown in Figure 2 consists of a simplified Transformer Encoder, MHFNet, a Transformer Decoder using the deformable self-attention mechanism, several feed-forward networks (FFN) for parameter prediction, and the Hungarian loss.

The encoder is composed of a single-layer Transformer Encoder structure, which consists of a multi-head self-attention module and a feed-forward network (FFN). It takes the output of the last layer of ResNet18 and reshapes the channels to 256, obtaining S5 with a feature scale of H/32 × W/32. The position information is encoded through sine embedding based on absolute positions and then calculated with the self-attention mechanism. The self-attention mechanism is as follows:(11)$$A=\mathrm{softmax}(\frac{{\mathrm{QK}}^{T}}{\sqrt{C}})\;,\;O=\mathrm{AV},$$where Q, K and V represent the sequences of queries, keys, and values obtained through a linear transformation of each input row. A represents the attention map, which measures nonlocal interactions to capture slender structures and global contexts. O represents the output of self-attention.

After that, the output $F_{5}$ is obtained by connecting to a residual layer with layer normalization following the FFN.

The decoder is composed of a multi-scale layer Transformer Decoder structure, which consists of a multi-head self-attention module, a multi-scale deformable attention module, and a feed-forward layer. It uses the mixed multi-scale features $\left\{S_{3},S_{4},F_{5}\right\}$ of MHFNet as another attention module inserted into each layer. The feature maps are of scales H/16 × W/16, H/32 × W/32 and H/64 × W/64, respectively, with the number of channels being 256. The input of the decoder is set as an empty N×C matrix, and all curve parameters are directly decoded at once. In addition, we also introduce learned lane embeddings of size N×C, which are called lane queries, serving as positional embeddings for implicitly learning global lane information. The attention mechanism uses the same formula (Equation (11)) as in the encoder. Sequentially, a decoding sequence of shape N×C is obtained, and this sequence is used for calculation in the multi-scale deformable attention module with the mixed multi-scale features output by MHFNet. The multi-scale deformable attention module is as follows:(12)$$MSDeformAttn(z_{q},{\hat{p}}_{q},{\left\{x^{l}\right\}}_{l=1}^{L})=\sum_{m=1}^{M}W_{m}\left[\sum_{l=1}^{L}\sum_{k=1}^{K}A_{mlqk}\cdot {W^{\prime }}_{m}x^{l}(\phi_{l}({\hat{p}}_{q})+\Delta p_{mlqk})\right],$$

${\left\{x^{l}\right\}}_{l=1}^{L}$ is the input multi-scale feature map, where $x^{l}\in R^{C\times H_{l}\times W_{l}}$. Let ${\hat{p}}_{q}\in {\left[0,1\right]}^{2}$ be the normalized coordinates of the reference point of each query element q, where m represents the attention head, L represents the input feature level, and k represents the sampling point. $\Delta p_{mlqk}$ and $A_{mlqk}$ represent the sampling offset and attention weight of the $k^{th}$ sampling point in the $l^{th}$ feature level and $m^{th}$ attention head, respectively. The scalar attention weights $A_{mlqk}$ are normalized by $\sum_{l=1}^{L}\sum_{k=1}^{K}A_{mlqk=1}$. We use normalized coordinates ${\hat{p}}_{q}$, where the normalized coordinates (0,0) and (1,1) represent the top-left and bottom-right corners of the image, respectively. The function $\phi_{l}({\hat{p}}_{q})$ in Equation (12) rescales the normalized coordinates ${\hat{p}}_{q}$ to the input feature map L.

Then, it is connected to a residual layer with layer normalization following the FFN. Then, they are independently decoded into lane parameters and category labels by feed-forward networks (FFNs) to obtain N final predictions.

The FFN prediction module uses three parts to generate a set of predicted curves H. A single linear operation directly projects the output of the decoder into N×2, and then a SoftMax layer operates on the last dimension to obtain the predicted labels (background or lane) $c_{i}$, i∈{1,…,N}. Meanwhile, a 3-layer perceptron with ReLU activation and hidden dimension C projects the output of the decoder N×4, where dimension 4 represents four sets of lane-specific parameters. Another 3-layer perceptron first projects the features into N×4 and then takes the average in the first dimension, thus generating four shared parameters.

### 3.6. Loss Function

The main difficulty in training this model lies in calculating the matching degree between the predicted lane parameters and the real lane. The proposed method uses the Hungarian fitting loss to perform bipartite matching between the predicted lane parameters and the real lane, and optimizes the regression loss for specific lanes based on the matching results.

First, the set of predicted lane parameters is defined as:(13)$$P={\left\{p_{i}\right\}}_{i=1}^{M}M,$$where M is defined to be greater than the number of lanes in common scenarios. Then, the set of real lane markings is represented as:(14)$$S={\left\{(c_{i},x_{i}^{\prime },y_{i}^{\prime })\right\}}_{i=1}^{M},$$

Then, by searching for the optimal injective function l:S→P, the bipartite matching problem between the set of predicted lane parameters and the set of real lane markings is defined as a problem of minimizing the cost:(15)$$\overset{\Lambda }{l}=\underset{l}{argmin}\sum_{i}^{M}L_{bmc}(P_{l_{i}},S_{i}),$$where $L_{bmc}$ is the matching cost between the $i^{\mathrm{th}}$ real lane and the predicted parameter set with index $l_{i}$. The matching cost takes into account both the category prediction and the similarity between the predicted parameter set and the real lane.

Finally, the regression loss function is defined as:(16)$$L=\sum_{i=0}^{M}-\mu_{1}\log\;g(c_{i})+Z(c_{i}=1)\mu_{2}L_{mae}(S_{i},S_{\overset{\Lambda }{l}(i)}),$$where $c_{i}$ is the probability of the category $g(c_{i})$, $S_{\overset{\Lambda }{l}(i)}$ is the fitted lane sequence, $\mu_{1}$, $\mu_{2}$ are the coefficients of the loss function, $L_{mae}$ is the mean absolute error, and Z(⋅) represents the indicator function.

## 4. Experiments and Analysis

The authors should discuss the results and how they can be interpreted from the perspective of previous studies and the working hypotheses. The findings and their implications should be discussed in the broadest context possible. Future research directions may also be highlighted.

### 4.1. Datasets

To comprehensively evaluate the proposed method, we conducted experiments on two representative lane-detection benchmarks: TuSimple [21] and CULane [4].

The TuSimple dataset is an autonomous driving dataset consisting of 6408 annotated images. These images were captured by a high-resolution (720 × 1280) forward-looking camera on US highways under different road conditions and weather conditions, both day and night. It specifically focuses on real highway scenarios and is composed of 3268 images for training, 358 images for validation, and 2782 images for testing. All images are of size 1280 × 720 pixels.

CULane is a large-scale open-source dataset collected by on-vehicle cameras in urban and highway scenarios in Beijing, China. It consists of 88,880 training images, 9675 validation images, and 34,680 test images. All images are of size 1640 × 5920 pixels. This dataset covers a variety of traffic scenarios, including normal, crowded, dazzling, shadow, no line, arrow, curve, cross-road, and night.

### 4.2. Evaluation Metrics

For the TuSimple dataset [21], there are three official metrics: False Positive Rate (FPR), False Negative Rate (FNR), and accuracy. We follow the literature and use the TuSimple metrics to calculate the accuracy. The prediction accuracy is calculated as follows:(17)$$accuracy=\frac{\sum_{clip}C_{clip}}{\sum_{clip}S_{clip}},$$where $S_{clip}$ is the number of real predicted points among the lane points in the video clip, and $C_{clip}$ is the number of correctly predicted lane points. A ground truth point is considered correctly predicted if there is a predicted point within 20 pixels of it, and if the accuracy is greater than 85%, it is considered a true positive. Otherwise, it will be regarded as a false positive (FP) or a false negative (FN).

For the CULane dataset [4], lanes are considered to be 30 pixels wide. If the intersection over union (IoU) between the prediction and the ground truth is greater than 0.5, the predicted lane is regarded as a true positive. We also use the false negative rate (FN) and the false positive rate (FP) to evaluate our method. Another evaluation metric we use is the F1 score, and its formula is as follows:(18)$$F1=\frac{2\times Precision\times Recall}{Precision+Recall},$$where $Precision=\frac{TP}{TP+FP}$, $Recall=\frac{TP}{TP+FN}$.

### 4.3. Experimental Parameters

The experimental environment built in this paper uses Ubuntu 20.04 as the operating system, PyTorch 1.13 as the deep learning framework, and Python 3.8 as the main development language. In terms of computing devices, GeForce RTX 4090 is selected. We use ResNet-18 and ResNet-34 as the backbone networks to create different versions of our proposed MHFS-Former. The input resolution for TuSimple is set to 640 × 360, and for CULane, it is set to 820 × 295.

In line with the common practices adopted in most contemporary experiments, we choose to augment the original data through methods such as random scaling, cropping, rotation, color jittering, and horizontal flipping. In the training parameter settings of the experimental model, ADAM is selected as the optimizer, and the learning rate is set to 0.0001. The learning rate is set to 0.0001 and decays by a factor of 10 every 450k iterations. The batch size is set to 16, and the loss coefficients $\omega_{1}$, $\omega_{2}$ and $\omega_{3}$ are set to 3, 5, and 2, respectively. The number of encoding layers and decoding layers is both set to 1. The fixed number N of predicted curves is set to 7, and the number of training iterations is set to 400 k.

### 4.4. Comparison Results

We present our results on the TuSimple dataset in Table 1 and provide a visualization of the experimental results in Figure 5. As shown in Table 1, our proposed method achieves the highest F1 score. We observe that CondLaneNet-ResNet18 [8] has the highest FNR score of 1.80%, but its FPR score is relatively poor at 6.17%. Similarly, SCNN [4] also has the highest FNR score of 1.80% and a relatively poor FPR score of 6.17%. In contrast, our ResNet18 version achieves a commendable FN score of 2.28%, and our FP score remains at a good level of 2.70%. Moreover, our ResNet34 version shows balanced performance, with an FP score of 2.34% and an FN score of 2.43%, indicating a strong balance between false positives and false negatives. The FPS of the ResNet18 version is 130, and the FPS of the ResNet34 version is 87, both of which meet the real-time detection requirements of lane lines. The FPS score of LSTR [17] is the highest, reaching 420.

Since the dataset focuses on presenting real highway scenarios recorded at different times of the day and night on US highways, the scenarios are relatively simple, and other methods have already achieved impressive results, but the performance gaps are small. However, our method still achieves a remarkable accuracy score of up to 96.88%, which is significantly better than other methods. The visualization results also demonstrate the robustness of the method. It is obvious from the figure that the lane predicted by our method almost perfectly matches the actual lane markings, proving the effectiveness of our proposed MHFNet in helping the Transformer network accurately locate the lane.

In Figure 6, the middle image in the first column shows a situation where vehicles block the lane at a distant turn; the bottom image in the second column shows that vehicles block the lane on the left side of the view; and the images in the third column show that lane cannot be fully seen due to vehicle occlusion. Since our proposed ShuffleLaneNet fully considers the information flow between feature channels and spatial positions, and our MHFS-Former can utilize the context information from the global environment, despite these visual obstacles, our method can still successfully and accurately identify the existence of lanes, demonstrating its strong robustness and accuracy.

The test results of our MHFS-Former on the CULane dataset and the comparison with other methods are shown in Table 2. As the table shows, the method we proposed significantly outperforms other methods. The R34 version of our MHFS-Former achieved an impressive 77.38% in the F1 score. Moreover, our method has demonstrated commendable performance under normal, glare, shadow, and no-line conditions. Our method reached an impressive 92.89% under normal conditions, 68.86% under dazzle conditions, 78.80% under shadow conditions, and 53.78% under no-line conditions outperforming other methods. This excellent performance indicates that MHFS-Former can effectively utilize the rich high-dimensional feature information embedded near the lane under complex-scene conditions. Therefore, our method significantly improves the accuracy and reliability of detecting and locating these lanes in various challenging and complex environments.

The outcomes of the tests conducted on the CULane dataset are illustrated in Figure 7. Our method can effectively detect and locate lanes in all scenarios. Obviously, in glare and shadow scenarios, the method we proposed can also accurately capture the positions of lanes, achieving a high level of detection and positioning, which demonstrates excellent adaptability and reliability. Since the ShuffleLaneNet fully considers the information flow between feature channels and spatial positions, and our MHFS-Former can skillfully utilize global information to accurately infer the occluded and hardly detectable lane, even in complex scenarios such as heavy vehicle congestion(crowd) and shadow occlusion, it can achieve high-precision lane detection, showing good detection results. In addition, specifically in other complex lane scenarios, our method does not perform poorly, which highlights the robustness and adaptability of our method in dealing with complex and challenging real-world driving situations.

### 4.5. Limitation and Discussion

Furthermore, it can be observed from Table 2 that Transformer-based models such as O2SFormer-ResNet18 and Laneformer-ResNet18 also perform well in lane detection on complex-scene datasets. O2SFormer-ResNet18 achieves an F1 score of 76.07%, second only to our MHFS-Former model, and demonstrates excellent performance under glare conditions, reaching 70.40%. Laneformer-ResNet18, with an F1 score of 71.71%, also proves its capability to handle complex scenarios.

Transformer-based models are generally effective at leveraging the rich high-dimensional feature information around lanes in complex scenes. However, the MHFS-Former, by integrating MHFNet and ShuffleLaneNet, goes beyond merely utilizing high-dimensional features. It delves deeper into the latent semantic relationships and hierarchical context information within these features. By extracting more abstract and discriminative representations, it can better distinguish lane characteristics from complex backgrounds. At the same time, through optimized network architectures and efficient computational mechanisms, it maintains excellent real-time performance. This enables our approach to significantly enhance the accuracy and reliability of lane detection and localization across diverse, challenging, and complex environments.

In the TuSimple dataset, the ResNet18 version of MHFS-Former has a frame rate (FPS) of 130, and the ResNet34 version has a frame rate of 87. Both versions meet the stringent requirements for real-time lane line detection in autonomous driving. In the context of self-driving systems, timely and accurate lane line recognition is the cornerstone of ensuring vehicle safety and enabling smooth path planning. A higher FPS not only means faster processing of visual information captured by vehicle-mounted cameras but also allows the autonomous driving system to respond more promptly to dynamic road changes, such as sudden lane shifts of surrounding vehicles or irregular lane markings. The fact that both versions of MHFS-Former achieve sufficient frame rates demonstrates their reliability in providing immediate feedback, which is crucial for preventing collisions and maintaining proper lane positioning. Moreover, these results highlight the effectiveness of our proposed method in balancing detection accuracy and computational efficiency, making it a viable solution for real-world autonomous driving applications where every millisecond of processing time can impact the overall safety and performance of the autonomous driving system.

### 4.6. Ablation Experiment Results

To validate the effectiveness of the proposed modules, we conducted an ablation study using the R18 version of the CULane dataset. The empirical results of these experiments are systematically presented in Table 3 to comprehensively evaluate the contributions and effectiveness of the proposed modules.

As shown in Table 3, in the base model, the F1 score was initially 77.11%. The adoption of ShuffleLaneNet led to an increase in the score of the base model to 77.16%, representing a significant improvement of 0.05%. ShuffleLaneNet significantly enhances the representational ability of the model by deeply exploring the channel and spatial information of lane features. Moreover, this module not only maintains the lightweight design of the model but also realizes the information interaction between features through channel shuffling operations, remarkably improving the detection performance of the model. Additionally, after adding MHFNet to the base model, the score rose to 77.30%, with a substantial increase of 0.19%. MHFNet enhances the effectiveness of multi-scale feature fusion by cascading multi-scale features, combining low-level features in the early stage with high-level features in the later stage. The introduction of adaptive spatial fusion operation alleviates the information conflict that may occur during the feature fusion process, further improving the effect of multi-scale feature fusion. When both the MHFNet and ShuffleLaneNet were added to the base model, the MHFS-Former model was derived, and there was a remarkable improvement in performance. The score soared to 77.38%, with a net gain of 0.27%, demonstrating its superior performance. Aiming at the problem that lane lines are vulnerable to interference in complex scenarios, MHFS-Former feeds multi-scale features into the Transformer Encoder to generate MHFNet. This network not only significantly enhances the feature representation ability of the model, accurately captures the linear structural features of lane lines and global context information, and realizes end-to-end detection without complex post-processing but also combines with ShuffleLaneNet. By deeply exploring the channel and spatial information of lane features and using the multi-reference deformable attention module to disperse the attention to the areas around the lane lines, it can better capture the global environmental information, enhance the perception ability of slender structures and global context, adaptively focus on the key features of lane lines, and effectively eliminate the interference of irrelevant background information.

## 5. Conclusions

In this work, we propose an end-to-end lane-detection network named MHFS-FORMER based on the Transformer. It innovatively takes multi-scale features as input information and introduces a novel multi-reference deformable attention module to distribute attention around objects, enabling better capture of the slender structure of lanes and the global environment while eliminating complex post-processing procedures. We also utilize MHFNet to enhance the mixed multi-scale features. By fusing the multi-scale features with the output of the Transformer Encoder, we obtain enhanced multi-scale features, thus boosting the effectiveness of multi-scale feature fusion. Furthermore, we have designed ShuffleLaneNet, which meticulously explores the channel and spatial information in lane features. By enhancing feature extraction, it improves the overall detection capability of the network and offers enhanced performance in diverse and complex scenarios. Our method achieved an accuracy score of 96.88% on the TuSimple dataset and an F1 score of 76.32% on the CULane dataset. The experimental results on both the CULane and TuSimple datasets have proven the effectiveness of this algorithm. In future research, we will keep making efforts to improve the detection accuracy in complex scenarios.

## Figures and Tables

**Figure 1 sensors-25-02876-f001:**
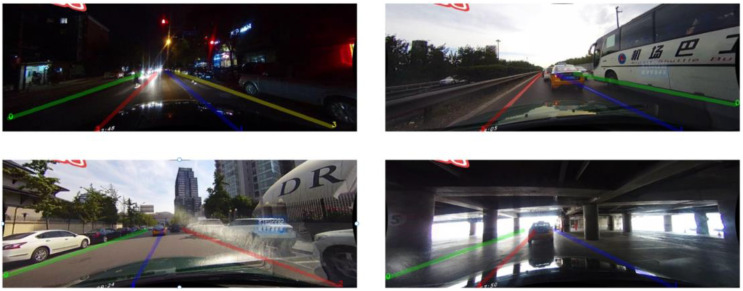
Several challenging lane-detection scenarios.

**Figure 2 sensors-25-02876-f002:**
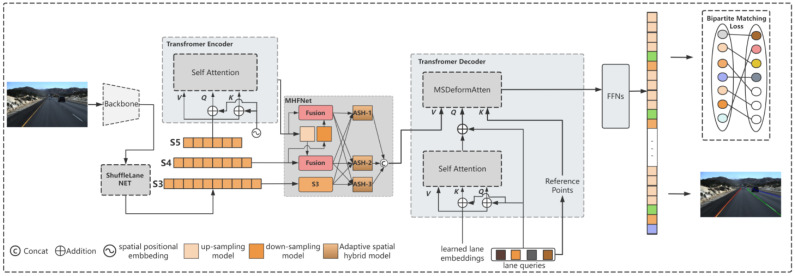
Overview of the lane-detection framework. It includes a CNN backbone for extracting feature maps from the input image, ShuffleLaneNet, MHNet, a lane-detection head that detects lanes through Transformer-based deformable convolution, and a bipartite matching loss for model training.

**Figure 3 sensors-25-02876-f003:**
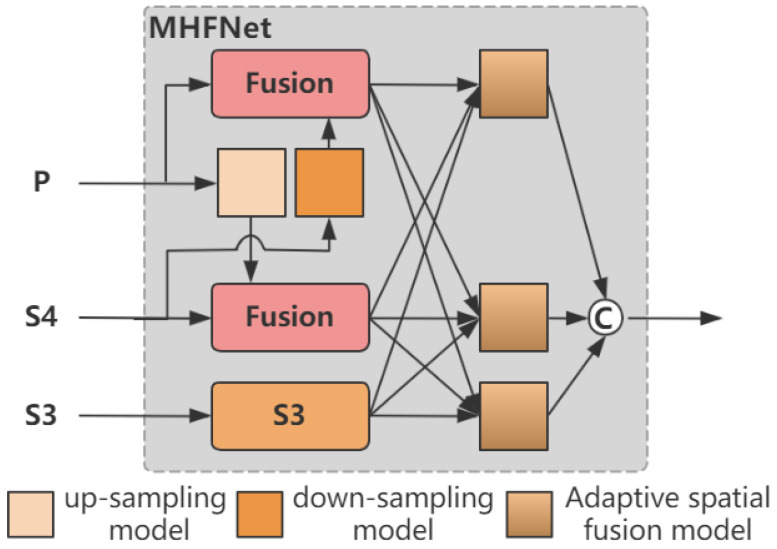
Network Structure of MHFNet.

**Figure 4 sensors-25-02876-f004:**
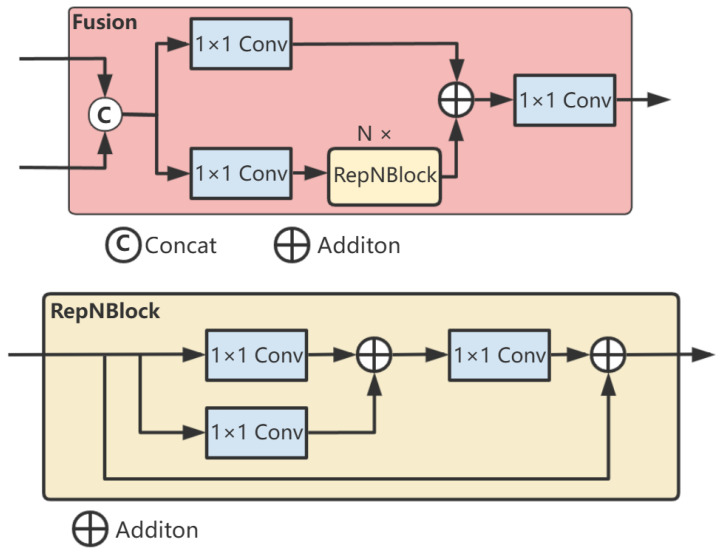
Network Structure of the Fusion Network.

**Figure 5 sensors-25-02876-f005:**
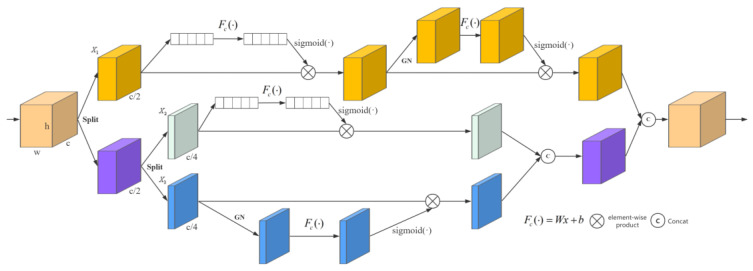
Structure of ShuffleLaneNet.

**Figure 6 sensors-25-02876-f006:**
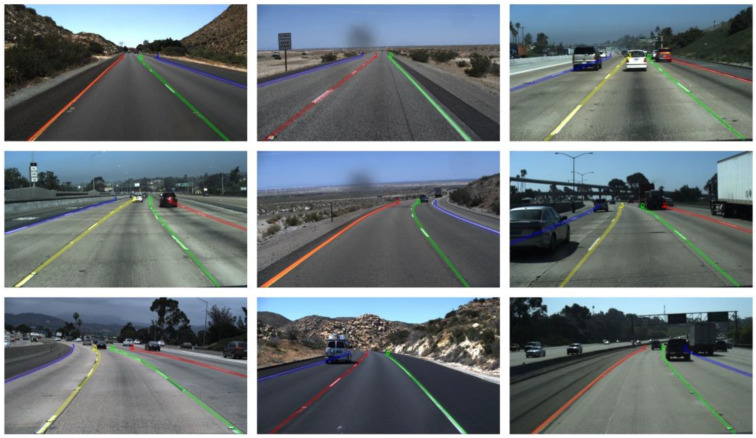
Visualization Results of Our MHFS-Former on the TuSimple Dataset.

**Figure 7 sensors-25-02876-f007:**
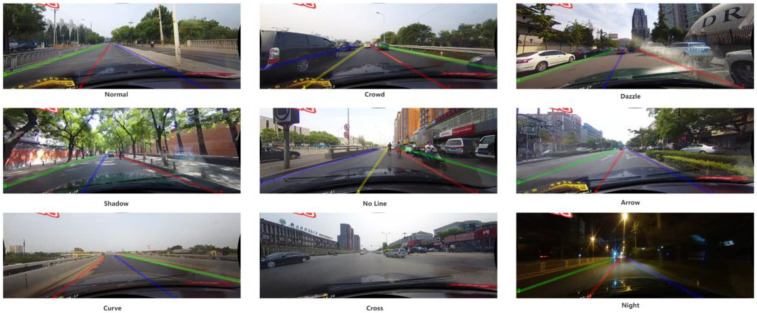
Visualization Results of Our MHFS-Former on the CULane Dataset.

**Table 1 sensors-25-02876-t001:** Comparison of Results on the TuSimple Dataset.

Method	F1	FPR	FNR	FPS
SCNN [4]	96.53	6.17	**1.80**	**7**
RESA-ResNet18 [6]	96.70	3.95	2.83	-
RESA-ResNet34 [6]	96.82	3.63	2.48	-
SAD-ResNet18 [7]	96.02	7.86	4.51	-
SAD-ResNet34 [7]	96.24	7.12	3.44	-
CondLaneNet-ResNet18 [8]	95.48	2.18	3.80	220
CondLaneNet-ResNet34 [8]	95.37	2.20	3.82	154
LaneATT-ResNet18 [11]	95.57	3.56	3.01	250
LaneATT-ResNet34 [11]	95.63	3.53	2.92	171
UFLDv2-ResNet18 [13]	95.65	3.06	4.61	312
UFLDv2-ResNet34 [13]	95.56	3.18	4.37	169
PolyLaneNet [14]	93.36	9.42	9.33	115
LSTR [17]	96.18	2.91	3.38	**420**
Eigenlanes [30]	95.62	3.20	3.99	-
BézierLaneNet-ResNet18 [22]	95.41	5.30	4.60	-
BézierLaneNet-ResNet34 [22]	96.65	5.10	3.90	-
MHFS-Former-ResNet18	96.42	**2.28**	2.70	130
MHFS-Former-ResNet34	**96.** **88**	2.34	2.43	87

The best results are in bold.

**Table 2 sensors-25-02876-t002:** Comparison of Results on the CULane Dataset.

Method	Normal	Crowd	Dazzle	Shadow	No Line	Arrow	Curve	Cross	Night	Total
SCNN [4]	90.60	69.70	58.50	66.90	43.40	84.10	64.40	1990	66.10	71.60
RESA-ResNet34 [6]	91.90	72.40	66.50	72.00	46.30	88.10	68.60	1896	69.80	74.50
SAD-ResNet18 [7]	89.80	68.10	59.80	67.50	42.50	83.90	65.50	1995	64.20	70.50
SAD-ResNet34 [7]	89.90	68.50	59.90	67.70	42.20	83.80	66.00	1960	64.60	70.70
LaneAF [9]	90.12	72.19	68.70	76.34	49.13	85.13	64.40	1934	68.67	74.24
UFLDv2-ResNet18 [13]	91.80	73.30	65.30	75.10	47.60	87.90	68.50	2075	70.70	75.00
UFLDv2-ResNet34 [13]	92.50	74.80	65.50	75.50	49.20	**88.80**	**70.10**	1910	70.80	76.00
BézierLaneNet-ResNet18 [22]	90.22	71.55	62.49	70.91	45.30	84.09	58.98	996	68.70	73.67
BézierLaneNet-ResNet34 [22]	91.59	73.20	69.20	76.74	48.05	87.16	62.45	888	69.90	75.57
Eigenlanes-ResNet18 [30]	91.50	74.80	69.70	72.30	51.10	87.70	62.00	1507	**71.40**	76.50
O2SFormer-ResNet18 [18]	91.89	73.86	**70.40**	74.84	49.83	86.08	68.68	2361	70.74	76.07
Laneformer-ResNet18 [31]	88.60	69.02	64.07	65.02	45.00	81.55	60.46	**25**	64.76	71.71
PINet(4H) [23]	90.30	72.30	66.30	68.40	49.80	83.70	65.60	1427	67.70	74.40
MHFS-Former-ResNet18	92.82	77.84	67.50	77.30	52.34	85.92	65.72	1223	69.63	76.83
MHFS-Former-ResNet34	**92.89**	**79.25**	68.86	**78.80**	**53.78**	86.70	67.70	1219	69.88	**77.38**

The best results are in bold.

**Table 3 sensors-25-02876-t003:** Ablation Experiments on the CULane Dataset.

Baseline	ShuffleLaneNet	MHFNet	F1
√			77.11
√	√		77.16
√		√	77.30
√	√	√	77.38

The "√" mark indicates method inclusion in each model’s ablation experiments.

## Data Availability

The data in this study are available upon request from the corresponding author.

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
