# Peer review of "MHFS-FORMER: Multiple-Scale Hybrid Features Transformer for Lane Detection"

_sensors, 2025, doi:10.3390/s25092876_

Round 1

Reviewer 1 Report

Comments and Suggestions for Authors

The paper proposes an MHFS-FORMER model inspired by the DETR architecture for lane detection. The novel aspect is the use of multi-scale features as input and the introduction of a multi-reference deformable attention module to distribute attention around objects. The manuscript is well-organized; however, there are some issues that need to be addressed:

(1) Given that lane detection is often performed in real-time scenarios, a discussion of the trade-offs between detection speed and accuracy should be included.

(2) The paper lacks a detailed explanation of why this specific combination results in such notable improvements.

(3) Traditional lane detection methods are quite mature. How does this model perform in complex scenarios (e.g., glare, shadow, no-line conditions)?

(4) Some of the literature used for accuracy comparisons are outdated. It would be advisable to incorporate more recent publications to enhance the comparative analysis.

Comments on the Quality of English Language

The english should be improved.

Author Response

For research article

Response to Reviewer 1 Comments

1. Summary

Thank you very much for taking the time to review this manuscript. Please find the detailed responses below and the corresponding revisions/corrections highlighted changes in the re-submitted files.

In response to the feedback regarding the need to enhance our English proficiency, we have meticulously revisited and refined the English formatting of our entire paper, with particular focus on the Introduction section. Additionally, based on the reviewers' comments, we have significantly augmented the experimental analysis section. Specifically, we have incorporated more comprehensive demonstrations of the model's real-time detection performance. We have also integrated the latest publications, especially those on lane detection algorithms leveraging Transformer models, thereby enriching the comparative analysis.To further strengthen the theoretical depth, we have added a new Section 4.5 dedicated to in-depth analysis of Transformer-based models and discussions on real-time lane detection.

2. Questions for General Evaluation

Reviewer’s Evaluation

Response and Revisions

Does the introduction provide sufficient background and include all relevant references?

Yes

Are all the cited references relevant to the research?

Yes

Is the research design appropriate?

Yes

Are the methods adequately described?

Can be improved

Enhanced the real-time detection indicators for improvement.

Are the results clearly presented?

Can be improved

Improvements were made by adding conclusion analysis to the ablation experiment section.

Are the conclusions supported by the results?

Can be improved

Improvements were made by adding recent publications to the comparative experiment section.

3. Point-by-point response to Comments and Suggestions for Authors

Comments 1: Given that lane detection is often performed in real-time scenarios, a discussion of the trade-offs between detection speed and accuracy should be included.

Response 1: Thank you for pointing this out. We agree with this comment. Therefore, we have already incorporated the real-time detection performance indicator, frames per second (fps), in Section 4.4. Moreover, in the newly added Section 4.5, we have further explored the relationship between real-time performance and lane detection algorithms.

Comments 2: The paper lacks a detailed explanation of why this specific combination results in such notable improvements.

Response 2: Agree. Accordingly, we have provided a detailed explanation of the significant improvements brought about by the specific combination in Section 4.6 to emphasize this point.

Comments 3: Traditional lane detection methods are quite mature. How does this model perform in complex scenarios (e.g., glare, shadow, no-line conditions)?

Response 3: Agree. In Section 4.4, we utilized two lane detection datasets to validate the performance of our model. The second dataset was specifically designed for complex scenarios. We conducted a dedicated analysis within this context and carried out comparative experiments. In these complex scenarios, our model achieved the best performance under conditions such as Normal, Crowd, Shadow, and No Line. Additionally, its performance in other scenarios is also quite robust.

Comments 4: Some of the literature used for accuracy comparisons are outdated. It would be advisable to incorporate more recent publications to enhance the comparative analysis.

Response 4: Thank you for pointing this out. We agree with this comment. Therefore, we have also integrated the latest publications, especially those on lane detection algorithms leveraging Transformer models, thereby enriching the comparative analysis.

4. Response to Comments on the Quality of English Language

Point 1: The english should be improved.

Response 1: Agree. In response to this feedback, we have thoroughly reviewed and polished the English language throughout the entire paper. Special attention was paid to the Introduction section, where we rephrased complex sentences, made the language more concise, and ensured better logical coherence.

5. Additional clarifications

Thank you again for your positive comments and valuable suggestions to improve the quality of our manuscript.

Reviewer 2 Report

Comments and Suggestions for Authors

The manuscript proposes an end-to-end Transformer-based lane detection model named MHFS-FORMER, which integrates MHFNet and ShuffleLaneNet. The model introduces a multi-reference deformable attention mechanism to better capture elongated structures and global context. Experiments on TuSimple and CULane datasets show competitive performance compared to prior methods. Reviewer has the following comments:

  1. While the proposed MHFS-FORMER achieves promising results, the methodological novelty remains unclear. Numerous recent studies have adopted Transformer-based architectures for end-to-end lane detection. It is recommended that the authors include a table to systematically compare MHFS-FORMER with existing Transformer-based lane detection methods, highlighting key differences in architecture, attention mechanisms, and feature fusion strategies.

  2. The comparative experiments lack sufficient details about whether training protocols (e.g., data augmentation, learning schedule, optimizer settings) are kept consistent across all methods. Please clarify these settings to ensure the fairness of the comparisons, particularly since some baselines may be sensitive to training conditions.

  3. Real-time lane detection in autonomous driving requires strong robustness under dynamic conditions. In particular, when the vehicle is at high speed or when a moving object (e.g., pedestrian or vehicle) crosses in front of the ego vehicle, stable and accurate lane detection is critical. The current paper focuses on static benchmarks. Please discuss how the proposed method would perform in such dynamic scenes, or consider adding evaluation/discussion in simulated or video-based settings.

Author Response

For research article

Response to Reviewer 2 Comments

1. Summary

Thank you very much for taking the time to review this manuscript. Please find the detailed responses below and the corresponding revisions/corrections highlighted changes in the re-submitted files.

In response to the feedback regarding the need to enhance our English proficiency, we have meticulously revisited and refined the English formatting of our entire paper, with particular focus on the Introduction section. Additionally, based on the reviewers' comments, we have significantly augmented the experimental analysis section. Specifically, we have incorporated more comprehensive demonstrations of the model's real-time detection performance. We have also integrated the latest publications, especially those on lane detection algorithms leveraging Transformer models, thereby enriching the comparative analysis.To further strengthen the theoretical depth, we have added a new Section 4.5 dedicated to in-depth analysis of Transformer-based models and discussions on real-time lane detection.

2. Questions for General Evaluation

Reviewer’s Evaluation

Response and Revisions

Does the introduction provide sufficient background and include all relevant references?

Can be improved

Improvements were made by adding an analysis of real-time performance.

Are all the cited references relevant to the research?

Yes

Is the research design appropriate?

Can be improved

Section 4.5 has been added to illustrate the advantages of the research design.

Are the methods adequately described?

Yes

Are the results clearly presented?

Can be improved

Improvements were made by adding conclusion analysis to the ablation experiment section.

Are the conclusions supported by the results?

Can be improved

Improvements were made by adding recent publications to the comparative experiment section.

3. Point-by-point response to Comments and Suggestions for Authors

Comments 1: While the proposed MHFS-FORMER achieves promising results, the methodological novelty remains unclear. Numerous recent studies have adopted Transformer-based architectures for end-to-end lane detection. It is recommended that the authors include a table to systematically compare MHFS-FORMER with existing Transformer-based lane detection methods, highlighting key differences in architecture, attention mechanisms, and feature fusion strategies.

Response 1: Thank you for pointing this out. We agree with this comment. Transformer-based lane detection methods excel in detecting complex scenes. The effectiveness of MHFS-FORMER can be verified through comparisons using the CULane dataset, which represents complex scenarios. However, simply adding a single Transformer-based lane detection method would seem redundant. Therefore, in the comparison on the CULane dataset, we have incorporated more recently developed Transformer-based lane detection methods. Moreover, to emphasize the differences, we have specifically added a comparison of Transformer-based lane detection methods in Section 4.5.

Comments 2: The comparative experiments lack sufficient details about whether training protocols (e.g., data augmentation, learning schedule, optimizer settings) are kept consistent across all methods. Please clarify these settings to ensure the fairness of the comparisons, particularly since some baselines may be sensitive to training conditions.

Response 2: Agree. Since most of the recent lane detection methods have employed nearly identical training protocols, we have specifically stated this point in Section 4.3 to clarify it.

Comments 3: Real-time lane detection in autonomous driving requires strong robustness under dynamic conditions. In particular, when the vehicle is at high speed or when a moving object (e.g., pedestrian or vehicle) crosses in front of the ego vehicle, stable and accurate lane detection is critical. The current paper focuses on static benchmarks. Please discuss how the proposed method would perform in such dynamic scenes, or consider adding evaluation/discussion in simulated or video-based settings.

Response 3: Thank you for pointing this out. We agree with this comment.Agree. Therefore, we have added the real-time detection indicator, frames per second (fps), in Section 4.4, and specifically discussed real-time lane detection in Section 4.5.

4. Response to Comments on the Quality of English Language

Point 1: The English is fine and does not require any improvement.

Response 1:  

5. Additional clarifications

Thank you again for your positive comments and valuable suggestions to improve the quality of our manuscript.

Round 2

Reviewer 1 Report

Comments and Suggestions for Authors

The authors have addressed all my queries. I recommend acceptance.